# Mitogenomics and Evolutionary History of Rodent Whipworms (*Trichuris* spp.) Originating from Three Biogeographic Regions

**DOI:** 10.3390/life11060540

**Published:** 2021-06-09

**Authors:** Jan Petružela, Alexis Ribas, Joëlle Goüy de Bellocq

**Affiliations:** 1Institute of Vertebrate Biology, Czech Academy of Sciences, Květná 8, 603 65 Brno, Czech Republic; joellegouy@gmail.com; 2Department of Botany and Zoology, Faculty of Science, Masaryk University, Kotlářská 2, 602 00 Brno, Czech Republic; 3Section of Parasitology, Department of Biology, Healthcare and the Environment, Faculty of Pharmacy and Food Sciences, University of Barcelona, 08007 Barcelona, Spain; aribas@ub.edu; 4Department of Zoology and Fisheries, Faculty of Agrobiology, Food and Natural Resources, Czech University of Life Sciences Prague, Kamýcká 129, 165 21 Prague, Czech Republic

**Keywords:** *Trichuris*, whipworms, mitogenomes, comparative genomics, rodents, palearctic, Afrotropical, Indomalayan, phylogenetics

## Abstract

*Trichuris* spp. is a widespread nematode which parasitizes a wide range of mammalian hosts including rodents, the most diverse mammalian order. However, genetic data on rodent whipworms are still scarce, with only one published whole genome (*Trichuris muris*) despite an increasing demand for whole genome data. We sequenced the whipworm mitogenomes from seven rodent hosts belonging to three biogeographic regions (Palearctic, Afrotropical, and Indomalayan), including three previously described species: *Trichuris cossoni*, *Trichuris*
*arvicolae,* and *Trichuris*
*mastomysi*. We assembled and annotated two complete and five almost complete mitogenomes (lacking only the long non-coding region) and performed comparative genomic and phylogenetic analyses. All the mitogenomes are circular, have the same organisation, and consist of 13 protein-coding, 2 rRNA, and 22 tRNA genes. The phylogenetic analysis supports geographical clustering of whipworm species and indicates that *T.* *mastomysi* found in Eastern Africa is able to infect multiple closely related rodent hosts. Our results are informative for species delimitation based on mitochondrial markers and could be further used in studies on phylogeny, phylogeography, and population genetics of rodent whipworms

## 1. Introduction

*Trichuris* is a genus of obligate parasitic nematode which infects a broad range of mammalian hosts, including humans. It is a common caecum parasite with fecal-oral transmission. Depending on the whipworm species in question, the number of hosts ranges from one to several taxonomically related species [1]. The overall degree of host specificity of *Trichuris* spp. remains an unresolved issue [2]. In humans, whipworms are responsible for the soil-transmitted disease trichuriasis, which manifests itself as gastrointestinal inflammation and can, with heavy infections, cause serious complications, such as rectal prolapse, iron-deficiency anemia, and long-term disabilities (especially in children). Trichuriasis affects over half a billion people annually and is considered to be one of the neglected tropical diseases burdening the developing world [3].

Rodents are the most diverse order of mammals, encompassing around 40% of all mammalian species, with family Muridae, native to the Old World, leading in both abundance and number of species [4]. Rodents are not only distributed worldwide but also have a rich evolutionary history with evidence of multiple adaptive radiation events [5,6]. They are also reservoirs for a plethora of parasitic and pathogenic organisms, some of which share a long evolutionary history with their rodent hosts, such as the yeast-like fungus *Pneumocystis* [7,8]. These facts make rodent parasites ideal for studies of host-parasite interactions and co-evolution. Such studies, however, require detailed genomic data.

Whipworms are common parasites of rodents, often showing prevalence over 20% in wild populations (e.g., 21.1% in *Mus musculus* in Central Europe [9], 23 and 38.5% in *Mastomys natalensis* in Africa [10,11], or 22% in *Sundamys muelleri* from Southeast Asia [12]. The mouse whipworm, *Trichuris muris*, is extensively used as an animal model of *T. trichiura*, the human whipworm [13], and is one of the three fully sequenced whipworm genomes together with *T. trichiura* [14] and *T. suis* [15].

Despite being one of the most recognized and studied nematodes, there is a lack of genomic data for murine *Trichuris* species, as the main interest has been with whipworms infecting humans, primates, and farm animals. Hence, there is a number of complete mitochondrial genomes in the literature for species such as sheep, *T. ovis* [16], or pigs, *T. suis* [17], but there is only one described mitochondrial genome for murine *Trichuris*, namely *T. muris* [18]. From these first mitogenomes, some peculiarities are already apparent. They exhibit a high degree of amino-acid divergence between species, unlike other nematode genera, such as *Ancylostoma* spp. [19,20] or *Toxocara* spp. [21]. They also have the ATP8 gene, which is absent in the rest of Nematoda (except early diverging *Trichinella*) [16,22]. As with other Nematoda, the whipworm mitogenomes are strongly AT biased.

Whipworm genetic data have been restricted to not only a few hosts but also restricted in geographic representation. This is unfortunate, since whipworms are often difficult to distinguish by morphology alone, and genetic data are helpful to reveal cryptic diversity [10,23,24]. In fact, molecular methods have become an increasingly important tool for completing the morphological description of different whipworm species, e.g., *T. arvicolae* [24], *T. mastomysi* and *T. carlieri* [10], *T. cutillasae* [25], or, most recently, *T. cossoni* [12]. Genetic studies based on nuclear markers, such as ITS-1, ITS-2, or 5.8S, have proven useful to distinguish closely related whipworm species [1,25,26]. However, the mitochondrial genome allows study of long non-recombining sequences with high substitution rates compared to nuclear. In addition, nematode mitogenomes are useful phylogenetic tools due to their diverse structure and gene-order rearrangements [22]. With the advent of next-generation sequencing, it has become increasingly cost-effective to sequence full mitogenomes. Because of this, multiple studies have now incorporated full mitogenomes to address the questions of whipworm phylogenetics or species delineation [16,17,27].

In this study, we characterised and performed a comparative genomic analysis of seven *Trichuris* mitochondrial genomes from different rodent species. Three of our samples are previously described whipworm species, *Trichuris mastomysi* [28], *Trichuris arvicolae* [24], and *Trichuris cossoni* [12]. Our whipworm samples cover three biogeographic regions (see Table 1): four species come from Afrotropical Muridae rodents, two species from Indomalayan Muridae rodents, and the remaining species from a Cricetidae rodent from the Palaearctic region. We also included the mitochondrial genome of the palearctic *T. muris* in our comparative analyses. The murid hosts sampled in our study came from three tribes (Murini, Rattini, and Praomyini—see Table 1). We then investigated the evolutionary history of the rodent *Trichuris* mitogenomes. The distribution of *Trichuris* samples in terms of rodent-host phylogeny and biogeographic origins allow us to evaluate which between these two factors plays the more dominant role in shaping the evolutionary history of rodent whipworms.

## 2. Materials and Methods

### 2.1. Sampling, DNA Extraction, and Sequencing

We used 7 *Trichuris* spp. (see Table 1 for sample information) preserved in 70% ethanol collected during previous studies on the evolutionary history, taxonomy, and ecology of rodents and their associated parasites and pathogens in Asia, Africa, and Europe [7,12,29]. Three of these samples were identified based on their morphological characteristics as *T. mastomysi*, recovered from the Natal multimammate mouse, *Mastomys natalensis,* from Tanzania; *T. arvicolae*, recovered from the common vole, *Microtus arvalis,* in the Czech Republic; and *T. cossoni,* recovered from the greater bandicoot rat, *Bandicota indica* in Laos [10,12,24]. DNA was extracted using Invisorb Spin Forensic kit (Stratec, Berlin, Germany) following the manufacturer instructions (but without using the carrier RNA). For 3 of our samples for which we obtained a very low amount of DNA after Qubit (Life Technologies, Eugene, OR, USA) measurement (<1 ng/μL), we performed a whole genome amplification (WGA) step using the Illustra Ready-To Go GenomiPhi V3 DNA amplification kit (GE Healthcare, Chalfont, UK) to increase the amount of DNA before library preparation. *Trichuris* DNA (WGA or not) was equimolarly pooled with DNA from another nematode belonging to the family Oxyuridae, and libraries were prepared using the KAPA HyperPrep kit (Roche, KAPA, Cape Town, South Africa). A pool of 24 dual indexed libraries (with 7 containing the *Trichuris* samples of this study and 17 additional ones from a wider nematode mitochondrial study) were sequenced using Illumina MiSeq platform and v2 chemistry (San Diego, CA, USA) (i.e., 2 × 250 bp paired-end reads) at the CEITEC Genomics Core Facility (Brno, Czech Republic). After de novo assembly of the mitogenomes, we found that the long non-coding region (NCR-L) was missing for all samples. So, for two samples for which we had some DNA left, i.e., *Trichuris* samples from *Mus caroli* and *Mus mahomet,* we characterised the variable region NCR-L by Sanger sequencing with primers designed in the surrounding NAD1 and NAD2 genes.

### 2.2. Genetic Characterisation of Rodent Hosts

For 2 host samples for which we had access to tissues, we confirmed the field identification by cytochrome b (CYTB) genotyping. We amplified the CYTB gene with primers H15915 and L14723 [30] using the Multiplex PCR kit (Qiagen, Hilden, Germany) in a final volume of 15 μL and using 1 μL of extracted DNA. Amplicons were Sanger-sequenced at Eurofins Genomics (Germany). For the Asiatic rodent samples, we did not have access to tissue or DNA, but a metagenomic study of the saliva sample from the same *Mus* specimen allowed us to retrieve a 300 nucleotide-long contig that BLASTed 100% with *Mus caroli* R5231 CYTB (GenBank AN: KJ530558). We thus used the complete CYTB from R5231 in the subsequent analysis. The CYTB of the *Mastomys natalensis* individual host of *T. mastomysi* was sequenced in a previous study (GenBank AN: MK454444) [31].

### 2.3. Mitogenome Assembly and Comparative Analyses

Obtained reads were quality checked using Fastqc software v0.11.5 [32] and trimmed using Skewer v0.2.2 [33]. The assembly was performed by Spades v3.12.0 [34] and the contigs corresponding to the mitochondrial genome identified with a pipeline consisting of Diamond v0.8.24.86 [35], Blast v2.5.0 [36], and BlobTools v0.9.19.4 [37] software. The two full mitogenome assemblies from *M. caroli* and *M. mahomet* whipworms were finalised in Geneious 11.1.5 after adding the sequences of the NCR-L variable region obtained by Sanger sequencing. Almost complete (N = 5) and complete (N = 2) assembled mitogenomes were analysed in Geneious and tools included therein. We added the mitogenome of *T. muris* (NC_028621) for the purpose of comparative genome analyses.

The sequences were aligned by Mauve genomic aligner [38]. The open reading frames were analysed with the Geneious ORF Tool using the invertebrate mitochondrial code and compared to the annotated genome of *T. muris*. Pairwise mean nucleotide and amino-acid identities were calculated for homologous genes. Putative secondary structures of 22 tRNA genes were identified using software ARWEN [39]. As an independent check, we compared these annotations to the ones obtained using the MITOS pipeline web server [40]. Additional statistical analysis was performed using a custom Python 3.6 script [41] Python Software Foundation, Python Language Reference, version 3.6, using packages Biopython 1.75 [42], Numpy 1.16 [43], and Matplotlib 3.3.4 [44]. We used a sliding-window approach with 1000 bp windows and 25 bp steps to compare GC contents and mean pairwise differences between each sample pair.

### 2.4. Phylogenetic Analysis

For the purpose of phylogenetic analysis, the mitogenomes of *T. ovis* (NC_018597), *T. trichiura* (NC_017750), and *Trichinella spiralis* (NC_002681) were added to the alignment and used as outgroup samples. jModelTest v2.1.10 [45] was used to test the substitution models. Phylogenetic analysis was performed via Bayesian inference (BI) conducted in MrBayes v3.2.6 [46] and complemented by a maximum likelihood phylogenetic analysis conducted in RaxML v8.2.12 [47]. The whole genome alignments were used for the phylogenetic analyses. GTR+G was used as substitution model. The analysis was performed using four independent Markov chain runs for 4,000,000 MCMC generations, sampling every 2000th generation. The first 25% of samples were discarded as burn-in. To compare the level of consistency between the two phylogenetic methods, we first checked the topologies for any inconsistencies and then plotted the individual values of pairwise differences between same sample pairs given by the different methods against each other. This comparison was performed to compare the difference between branch lengths as given by the two methods of phylogenetic analysis.

Only the Bayesian inference analysis using MrBayes was performed for the host samples. We used similar settings for host CYTB sequence analysis, with *Rhizomys pruinosus* (MH189045) used as outgroup. For host species for which we did not have access to tissue, we used the following GenBank sequences: *Bandicota indica*: HM217476, *Microtus arvalis*: KX380038, *Mus musculus*: KX790793, and *Mus mahomet*: MN223610. The only host/parasite difference in the MrBayes settings was the choice of JC96 as the host substitution model.

## 3. Results

### 3.1. Genomic Features and Annotations

All assemblies were cut in the NCR-L, which uniformly had the lowest coverage. We completed this region for sample LO613 obtained from *Mus caroli* and ETH232 obtained from *Mus mahomet* via Sanger sequencing. However, the NCR-L was missing from the other assembled sequences. The two complete mitogenomes, *T.* sp. ex *Mus caroli* and *T.* sp. ex *Mus mahomet* (Figure 1), had lengths 14,165 bp and 14,100 bp, respectively (see Table 2 for comparison and Appendix A for comparison of the incomplete genomes). All mitogenomes contained 13 protein coding genes (PCGs) (COX1-COX3, NAD1-NAD6, NAD4L, ATP6, ATP8, and CYTB), with NAD5 being the longest and ATP8 the shortest in all samples, 2 rRNA genes, and 22 tRNA genes (except *T. cossoni*, see below). Four PCGs (NAD2, NAD5, NAD4, and NAD4L) and 10 tRNA genes (tRNA-Met, tRNA-Phe, tRNA-His, tRNA-Arg, tRNA-Pro, tRNA-Trp, tRNA-Ile, tRNA-Gly, tRNA-Cys, and tRNA-Tyr) were encoded by the L-strand, with the rest being encoded by the H-strand. The NCR-L was located between NAD1 and tRNA-Lys, while the short non-coding region (NCR-S) was found between NAD3 and tRNA-Ser(UCN) (Figure 1). These findings are consistent with published literature on *Trichuris* mitogenomes. Genes NAD1 and NAD2 were incomplete, and the tRNA-Lys sequence was missing in *T. cossoni* ex *Bandicota indica*, as these lie in the immediate vicinity of the missing NCR-L region. In the same sample, the NAD4L gene was abnormally longer (336) than the rest (249–252 bp), with an overlap with NAD4 gene. The reason is a mutation in position 7427 with A replacing T and changing the function of an original stop codon. We checked the trimmed reads to rule out an assembly error. For the seven studied mitogenomes, the average nucleotide frequencies were 34.6% A, 12.3% C, 13.9% G, and 37.2% T. The nucleotide composition was strongly biased towards A+T (73.3%), with T being the most favoured and C being the least favoured nucleotide.

All PCGs used ATG, ATA, TTG, or ATT as their initiation codon. All PCGs except for NAD5 had complete termination codons. In the whole dataset, only COX2 and COX3 (and possibly NAD4, but this could not be inferred due to the missing sequence in *T. cossoni* sample) had initiation and termination codons identical across all samples. Most of the 22 tRNA genes could be folded into standard secondary structures. The length of tRNA genes ranged from 45 to 94 bp, consistent with other literature on *Trichuris* mitogenomes, and the tRNA-Ser lacked D-arm and loop [16]. In each genome, the 12S-rRNA gene was located between tRNA-Ser(AGN) and tRNA-Val, while 16S-rRNA gene was found between tRNA-Val and ATP6 gene. There were no differences in tRNA anticodons across the samples. The position of NCR-L could be inferred from the assembly even in those samples, being between the genes NAD1 and tRNA-Lys. However, very limited conclusions about the base composition in said region could be drawn, since the coverage of NCR-L was either too low or non-existent in the five samples mentioned. The NCR-S was located between NAD3 and tRNA-SerUCN.

### 3.2. Comparative Genomic Analyses

The reference genome of *T. muris* was added to our samples for the analyses within this section. For summary statistics, see Table 3. The sequence divergences between the genomes were substantial, ranging from 2.2% to 34.5%, with average pairwise divergence being 21.7% (Figure 2). The number and order of mitochondrial genes (13 PCGs, 22 tRNA genes, and 2 rRNA genes) and non-coding regions was identical across all studied mitogenomes. The mean pairwise divergence between the samples was 24.7%, with 7320 invariant sites across the whole dataset (45.8%). The magnitude of nucleotide sequence divergence in each gene ranged from 15% in COX1, being the most conserved gene, to 32.8% in ATP8, being the least conserved one. Amino-acid sequences were also compared. There were a total of 1361 amino-acid substitutions across all individual proteins without NAD1 and NAD2. The mean pairwise difference ranged from 7.3%–38%, with COX1 being the most and ATP8 the least conserved protein. There were no marked differences in GC content across our samples (Figure 3). Both non-coding regions were heavily AT biased (AT percentages 78.6% for NCR-L and 80.2% for NCR-S). Although the information about NCR-L came from two samples only, these were very similar to the NCR-L of *T*. *muris* both in length (82 bp in *T*. sp. ex *Mus caroli* and 100 bp in *T.* sp. ex *Mus mahomet* vs. 112 bp in *T*. *muris*) and composition (GC content 21.4% average in our samples vs. 19.6% in *T. muris*). The average length of NCR-S (107.4 bp in our samples vs. 106 bp in *T. muris*) and GC content was similar to the *T. muris* (19.8% average in our samples vs. 20.8% in *T. muris*).

### 3.3. Phylogenetic Analysis

Both maximum likelihood and Bayesian inference analyses gave the same phylogenetic topology (Figure 4) and very similar branch lengths. The difference between individual pairwise divergences for each method of phylogenetic analysis was low, ranging from 0–0.25, with average difference of 0.056 (for more details, see Appendix A). The phylogenetic analyses revealed that the *Trichuris* derived from Afrotropical rodents clustered into a single haplogroup (Figure 4). Samples from this haplogroup had a mean pairwise difference at 4.4%, with 12,917 sites being completely invariant (91.5%). *Trichuris* from Palearctic rodents, *M. arvalis* (Czech Republic) and *M. musculus,* formed a sister group to this haplogroup. *T.* sp. ex *Mus caroli* (Laos) was basal to all other rodent derived *Trichuris,* with *T. cossoni* ex *Bandicota indica* (Laos) being the second most basal sample. The samples from Afrotropical and Palearctic regions formed two highly supported monophyletic haplogroups, while the Indomalayan samples did not form a monophyletic group. The divergence (per Bayesian inference analysis) between whipworms found in ungulates (*T. ovis*) and whipworms found in rodents was of the same order of magnitude than between two rodent pinworms: The pairwise difference between *T.* sp. ex *Mus caroli* and *T. muris* was 33.1%, while the difference between it and the *T. ovis* was 36.7%. The phylogenetic pattern strongly supports geographical clustering of rodent-derived *Trichuris,* with no clear evidence of codivergence between the host and parasite. The reconstructed host phylogeny (see Appendix A), although partially unresolved, agrees with published results on murine phylogenetics [5,6,48]. Individual phylogenetic distances are given in Table 4.

## 4. Discussion

We have described seven mitochondrial genomes from different rodent whipworms. Two genomes were described completely, while in five others, the information about NCR-L is lacking. The divergence between the genomes was substantial (average pairwise divergence of 21.7%). The gene organisation was identical in all samples and did not differ from that of *Trichuris* found in other animals. The missing tRNA-Lys sequence in *T. cossoni* was a result of sequencing issues, and there is no reason to suspect that the gene itself is not present within the said mitogenome. GC content did not differ drastically across the samples (Figure 2), and the degree of genome conservation (Figure 3) is in line with other studies, with rRNA subunits being relatively the most conserved region [16,17].

Previously, a high degree of amino-acid divergence between different *Trichuris* species was reported [16,17] relatively to other nematodes. In accord with these findings, we observed a similar pattern, with mean pairwise amino-acid divergence at the least conserved gene (ATP8) being 38%, which is similar to the reported divergence between *T. ovis* and *T. discolor* (33.9%). When the samples *Trichuris* ex *Mus caroli* and *Trichuris* ex *Bandicota indica* were excluded from the comparison, this number dropped to 18.1%, as these two samples are the most distinct of the whole dataset. However, this number is still high, since a divergence of around 10% is common between different nematode species, such as in the genera *Ancylostoma* with around 4% [19,20] or *Toxocara* spp. with around 5–7% [21]. We suggest these observations are likely not the result of an unusually high mtDNA substitution rates within the *Trichuris* genus but rather due to sparse sampling from deep mitogenome phylogeny. Given that enoplids are a group that diverged early in the evolution of Nematoda, e.g., the related genus *Trichinella* diverged from other Nematoda as early as 275 million years ago [49,50], it is probable that the reported amino-acid divergences are a function of deep time: whipworms are likely also an early diverging nematode clade. If so, whipworms may contain a great cryptic biodiversity only starting to be discovered, which is suggested by the recent studies as well as by our own data. Extrapolating from previous studies that compared the amino-acid differences between *Trichuris* species [16,51], the pairwise amino acid divergences we observe would suggest four of our seven mitogenomes comes from different whipworm species. A fifth taxon of whipworms from African murid genera *Mastomys* and *Mus* rodents is also clear. The issue of species delimitation within this taxon will be returned to below.

Our results also support a clear geographic pattern of differentiation between different whipworm lineages, with no evidence of co-divergence between the hosts and parasites in question. The samples clustered together based on the continent/area of their origin: the two samples from Ethiopia, although sampled from the Murini and Praomyini tribes, cluster together before clustering with two other *Trichuris* samples from the *Praomyini* tribe from Tanzania and Kenya. Similarly, *T. arvicolae* from a Cricetidae rodent host clusters first with *T. muris* found in *Apodemus sylvaticus*, *Mus musculus,* or *Rattus rattus* in Europe [2,52], those hosts belonging to the Murini and Rattini tribes of the Muridae family. This pattern might be expected in parasites without strict host specificity. The samples from Asian rodents (*Mus caroli* and *Bandicota indica*) did not form a single monophyletic group. In fact, *Trichuris* genomes obtained from these two samples were the most distinct pair over all the studied samples (pairwise phylogenetic divergence being 34.5%), strongly suggesting an existence of two distinct *Trichuris* haplogroups in Southeast Asia. This is in line with recent results of Ribas et al. [12], which showed that *Trichuris* from Asian rodents are distinct from the population of European *Trichuris* based on morphological and genetic characteristics. Our phylogenetic analysis, while only preliminary, also hints at Southeast Asia as a possible evolutionary origin of murine *Trichuris*, as the samples obtained there both clustered basally and were as distinct from each other as they were from the samples from Africa and Europe.

Our study also has potential implications for the question of species delineation. In our phylogenetic analysis, we used two recognized species, *T. muris* and *T. arvicolae* (described on the basis of morphological differences), with 17.7% sequence divergence. Yet several other samples, such as *T. cossoni* ex *Bandicota indica* or *Mus caroli,* were in fact more divergent from any other sample than the aforementioned figure. The average distance of *Trichuris* ex *M. caroli* and *B. indica* to other murine samples was 33.4% and 29.8%, respectively (see Table 3 for more details). These numbers are similar to the phylogenetic distance between two recognized and distinct species, *T. muris* and *T. ovis*, which in the same analysis amounts to 35.4%. This is in line with results obtained by Liu et al. [16], who found that the genome divergence between *T. ovis* and *T. discolor* was 21.9%. If one were to go by the average distance between recognized *Trichuris* species, one ought to already recognize some of our samples as a putative species.

Returning to the African rodent-host samples: We can state that our observations for samples obtained from species *Mastomys natalensis*, *Mastomys erythroleucus*, *Praomys missonei,* and *Mus mahomet* are consistent with them all belonging to a single species: *Trichuris mastomysi*. These samples not only formed a monophyletic haplogroup in our phylogenetic analysis, but they also exhibit low mean pairwise distance (4.7%), while being strongly distinct from all other samples (Figure 4). It should be emphasized this consistency does not preclude the three samples being representatives of more than one species. The answer to that question depends on how species are defined. Most species definitions would require data from more than one locus to resolve species-hood (the mitogenome is assumed to comprise a single non-recombining locus). *Trichuris mastomysi* has been already reported on another species than *Mastomys natalensis*, namely *Mastomys erythroleucus* in Western Africa [53]. In this study, *T. mastomysi* was described as clade II, while another clade, clade I, grouped together *Trichuris* individuals found in three species of the genus *Mastomys (M. natalensis, M. erythroleucus,* and *M. huberti), Arvicanthis niloticus,* and *Gerbilliscus gambianus.* The genetic distance separating clades I and II was 7.5% based on the nuclear ITS-1,5.8S and ITS-2 rDNA genes, suggesting clade I corresponds to individuals belonging to a different *Trichuris* species than *T. mastomysi*. It thus appears that, whatever *T. mastomysi* or *Trichuris* clade I, both species have the ability to infect several related murid-host species. Nevertheless, it cannot be conclusively ruled out that *M. natalensis* is the main host of both *Trichuris* species and the other hosts are just the result of spillovers and thus accidental in nature. There are two important considerations regarding this issue. First, there has never been a systematic investigation into *Trichuris* on the aforementioned species, so the expected prevalence in African rodents studied here is unknown. Second, the more rodent species *T. mastomysi* is found to inhabit, the more likely are two specific scenarios. Either *T. mastomysi* naturally occurs in multiple related rodent species and can infect these without difficulty, or it is particularly prone to spillover events. More systematic research into prevalence of *T. mastomysi* in different rodent species is needed before this question can be satisfactorily answered.

## 5. Conclusions

In the present study, we described seven new mitogenomes of rodent *Trichuris* representing four different species and analysed their phylogenetic relationships. It would be interesting to get additional material of *Trichuris* ex *Mus caroli* to check if morphological characters, generally in males, support the large genetic divergence observed at the mitochondrial level. The large divergence between our *Trichuris* species coming from three major biogeographic zones in the evolutionary history of the Muridae provides an important source of information for mitochondrial markers to be used as basis for subsequent rodent-borne *Trichuris* phylogenetic, phylogeographic, and population genetic studies.

## Figures and Tables

**Figure 1 life-11-00540-f001:**
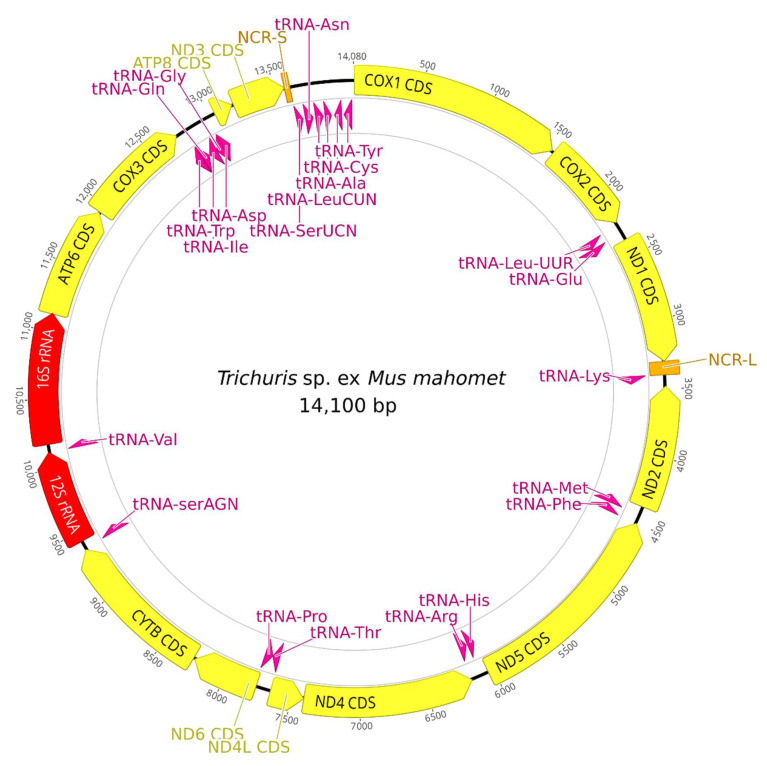
Structure of mitogenome for *T.* sp. ex *Mus mahomet*.

**Figure 2 life-11-00540-f002:**
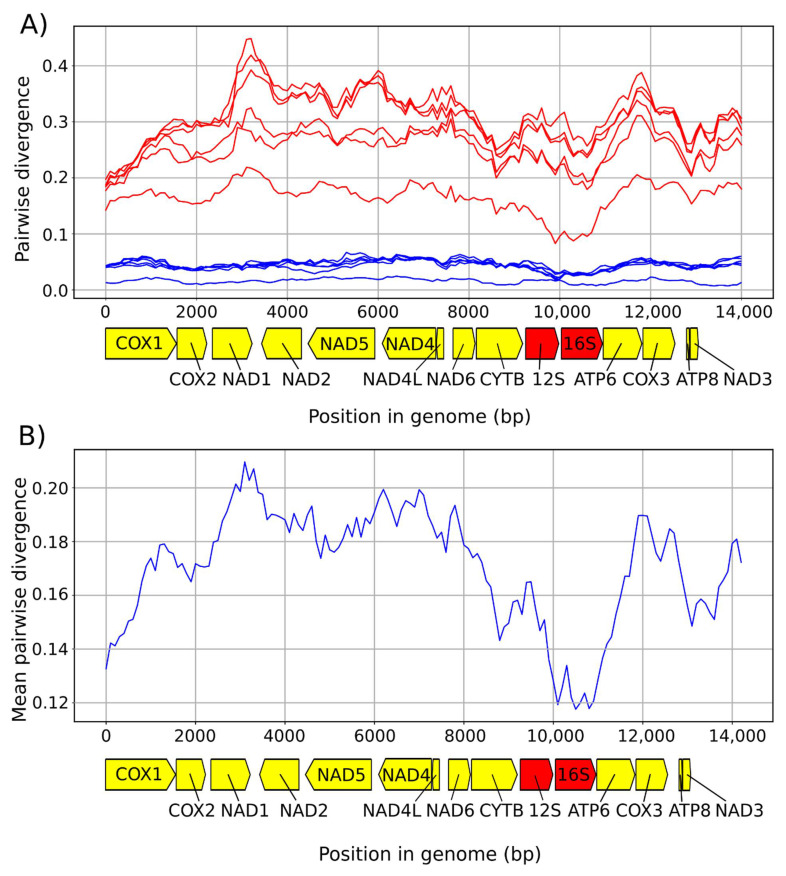
(**A**) Individual pairwise distances for all samples measured by using a sliding window of 1000 bp with 100 bp steps. The blue lines represent the distances among samples from the Afrotropical region only; the red lines represent the rest of the cal region and samples from elsewhere. (**B**) Mean pairwise distance for all samples measured by using a sliding window of 1000 bp with 100 bp steps.

**Figure 3 life-11-00540-f003:**
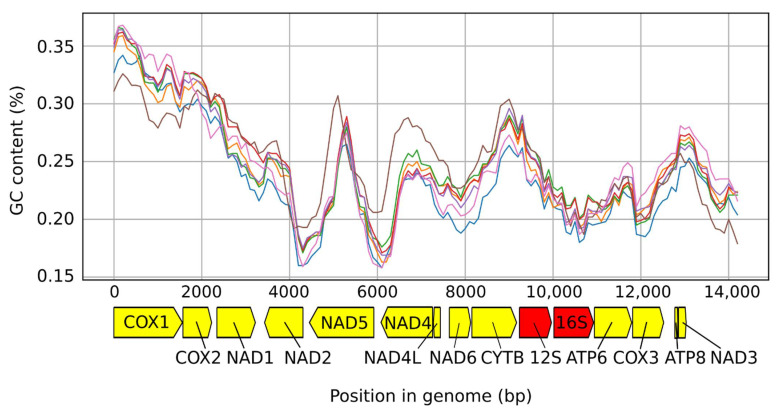
Mean GC content for all samples measured by using a sliding window of 1000 bp with 100 bp steps.

**Figure 4 life-11-00540-f004:**
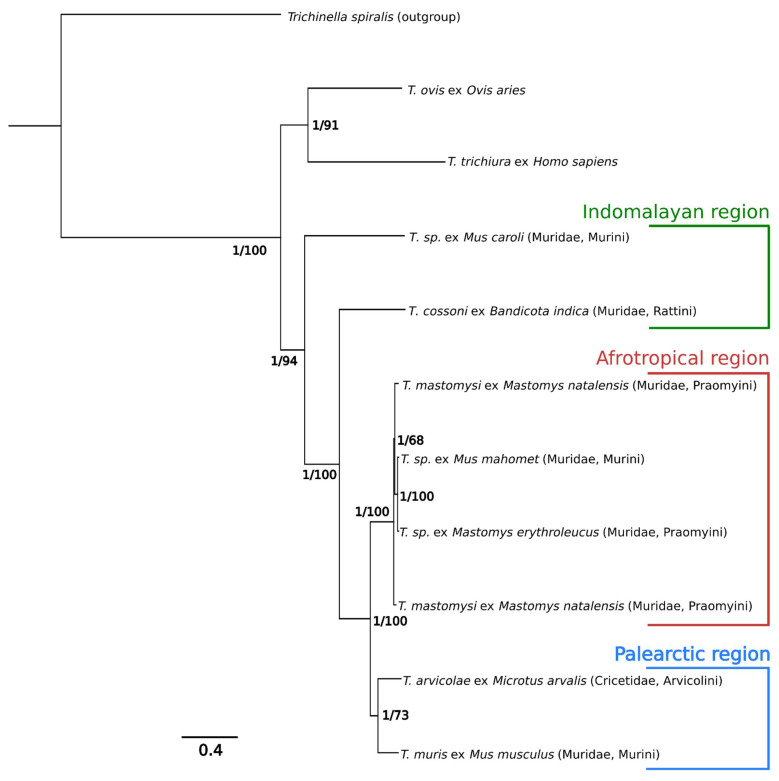
Phylogenetic tree based on whole mitogenome alignment of all *Trichuris* samples used in our study, with the addition of *T. trichiura* and *T. ovis*. *Trichinella spiralis* used as an outgroup. The node labels denote Bayesian posterior probabilities (BPP) followed by ML bootstrap values.

**Table 1 life-11-00540-t001:** Samples and host organisms used in the present study. Abbreviations: Gb. AN, GenBank accession number; Lat., latitude; Lon., longitude.

*Trichuris* Species	Host	Rodent Family/Tribe	Biogeographic Region	Gb. AN	Gb. AN (Host)	Year	Country	Lat.	Lon.
*Trichuris arvicolae*	*Microtus arvalis*	Cricetidae/Arvicolini	Palaearctic	MZ229684	KX380038	2014	Czech Republic	50.1327	12.6156
*Trichuris* sp.	*Praomys misonnei*	Muridae/Praomyni	Afrotropical	MZ229689	MZ222133	2010	Kenya	0.2382	34.8647
*Trichuris* sp.	*Mastomys erythroleucus*	Muridae/Praomyni	Afrotropical	MZ229686	MZ222134	2012	Ethiopia	4.9624	38.2068
*Trichuris* sp.	*Mus mahomet*	Muridae/Murini	Afrotropical	MZ229688	MN223610	2012	Ethiopia	6.8638	37.7629
*Trichuris mastomysi*	*Mastomys natalensis*	Muridae/Praomyni	Afrotropical	MZ229690	MK454444	2016	Tanzania	-8.0383	36.8444
*Trichuris* sp.	*Mus caroli*	Muridae/Murini	Indomalayan	MZ229687	KJ530558	2016	Lao PDR	18.3701	102.4998
*Trichuris cossoni*	*Bandicota indica*	Muridae/Rattini	Indomalayan	MZ229685	HM217476	2016	Lao PDR	18.3412	102.4782

**Table 2 life-11-00540-t002:** Mitogenome organisation of *T.* sp. ex *Mus caroli* and *T.* sp. ex *Mus mahomet*. Abbreviations: ini, initiation; ter, termination.

Gene/Region	Position in Genome				Strand	Ini/Ter Codons		Anticodon
	*T.* sp. ex *Mus mahomet*		*T.* sp. ex *Mus caroli*			*T.* sp. ex *Mus mahomet*	*T*. sp. ex *Mus caroli*	
	Position	Length	Position	Length				
COX1	1–1545	1545	1–1545	1545	H	ATG/TAG	ATG/TAA	
COX2	1562–2239	678	1556–2236	681	H	ATG/TAA		
tRNA-Leu-UUR	2253–2317	65	2258–2326	69	H			TAA
tRNA-Glu	2321–2383	63	2349–2414	66	H			TTC
NAD1	2401–3303	903	2428–3330	903	H	ATT/TAG	ATA/TAG	
NCR-L	3304–3403	100	3331–3412	82				
tRNA-Lys	3404–3466	63	3413–3472	60	H			TTT
NAD2	3484–4377	894	3485–4378	894	L	ATA/TAA		
tRNA-Met	4378–4439	62	4379–4442	64	L			CAT
tRNA-Phe	4450–4506	57	4447–4506	60	L			GAA
NAD5	4507–6043	1537	4520–6059	1542	L	ATC/TAA	ATA/TAA	
tRNA-His	6044–6100	57	6060–6115	56	L			GTG
tRNA-Arg	6111–6172	62	6130–6196	67	L			TCG
NAD4	6171–7406	1236	6195–7424	1230	L	ATG/TAA	ATT/TAA	
NAD4L	7406–7654	249	7451–7702	252	L	ATT/TAA		
tRNA-Thr	7656–7712	57	7705–7757	53	H			TGT
tRNA-Pro	7714–7772	59	7763–7818	56	L			TGG
NAD6	7765–8247	483	7811–8286	476	H	ATA/TAA	ATT/TAA	
CYTB	8258–9370	1113	8312–9424	1113	H	ATG/TAA	ATG/TAG	
tRNA-serAGN	9381–9436	56	9430–9484	55	H			TCT
12S rRNA	9437–10,119	683	9484–10,193	710	H			
tRNA-Val	10,121–10,173	53	10,193–10,248	56	H			TAC
16S rRNA	10,176–11,128	953	10,251–11,212	962	H			
ATP6	11,126–11,935	811	11,222–12,037	816	H	ATA/TAA	ATG/TAG	
COX3	11,939–12,718	780	12,044–12,829	786	H	ATG/TAA		
tRNA-Trp	12,723–12,788	66	12,829–12,890	62	L			TCA
tRNA-Gln	12,792–12,847	56	12,898–12,952	55	H			TTG
tRNA-Ile	12,855–12,919	65	12,959–13,023	65	L			GAT
tRNA-Gly	12,920–12,976	57	13,031–13,088	58	L			TCC
tRNA-Asp	12,977–13,037	61	13,098–13,154	57	H			GTC
ATP8	13,038–13,178	141	13,155–13,307	153	H	ATA/TAG	ATT/TAA	
NAD3	13,187–13,528	342	13,307–13,648	342	H	ATT/TAG	ATT/TAA	
NCR-S	13,529–13,635	107	13,649–13,761	113				
tRNA-SerUCN	13,636–13,688	53	13,762–13,815	54	H			TGA
tRNA-Asn	13,706–13,766	61	13,814–13,882	69	H			GTT
tRNA-LeuCUN	13,788–13,853	66	13,889–13,951	63	H			TAG
tRNA-Ala	13,865–13,919	55	13,967–14,020	54	H			TGC
tRNA-Cys	13,945–14,001	57	14,030–14,088	59	L			GCA
tRNA-Tyr	14,020–14,076	57	14,098–14,155	58	L			GTA

**Table 3 life-11-00540-t003:** Summary of differences in nucleotide sequences and predicted amino-acid sequences. * statistics for NAD1 and NAD2 were computed excluding the sample of *T. cossoni*, which had incomplete sequences at these sites. Abbreviations: Nuc, nucleotide; ini, initiation; ter, termination. ** statistics for NCR-L were computed using only samples of *T.* sp. ex *Mus mahomet* and *T.* sp. ex *Mus caroli*.

Gene/Region	% of Invariant Sites (nuc)	Mean Pairwise diff. (nuc)	Mean Pairwise Difference (aa)	% GC	Mean Nucleotide Length	Ini/Ter Codons
COX1	66.7	15.00	7.3	33.4	1546	variable
COX2	58.3	18.60	15.4	32.4	678	ATG/TAA
NAD1 *	59.5	15.5	17.7	36.8	903	variable
NCR-L **	56.0	44.0	-	21.4	96.5	-
NAD2 *	54	17.3	24.3	37.1	894	ATA/TAA
NAD5	59.4	17.80	24.5	25.3	1538	variable
NAD4	48.8	22.40	21.3	24.6	1203	variable
NAD4L	51.8	20.90	24	25.7	250	variable
NAD6	45.8	24.50	29.2	23.9	481	variable
CYTB	59.5	17.70	16.5	28.6	1116	variable
12S rRNA	57.9	17.20	-	24.2	688	-
16S rRNA	55.6	19.70	-	22.7	953	-
ATP6	45.0	24.10	28	22.4	809	variable
COX3	52.8	20.60	15.4	28.1	782	ATG/TAA
ATP8	31.3	32.30	38	23.7	143	variable
NAD3	60.8	16.8	16.6	22.6	342	variable
NCR-S	17.6	38.4	-	19.8	107.4	-

**Table 4 life-11-00540-t004:** Phylogenetic distance matrix of all samples used in the Bayesian inference analysis excluding the outgroup sequence of *Trichinella spiralis*.

	***T.*** **sp. ex *M. caroli***	***T. mastomysi***	***T.*** **sp. ex *P. missonei***	***T.*** **sp. ex *M. mahomet***	***T.*** **sp. ex *M. erythroleucus***	***T. arvicolae***	***T. muris***	***T. cossoni***	***T. ovis***	***T. trichiura***
***T.*** **sp. ex *M. caroli***		33.19	33.33	33.43	33.46	32.66	33.14	34.48	36.8	38.52
***T. mastomysi***	33.19		5.09	5.35	5.4	19.85	19.92	29.21	35.46	37.61
***T.*** **sp. ex *P. missonei***	33.33	5.09		5.21	5.14	19.84	20.01	29.06	35.55	37.8
***T.*** **sp. ex *M. mahomet***	33.43	5.34	5.21		2.16	20.1	19.94	29.28	35.65	37.96
***T.*** **sp. ex *M. erythroleucus***	33.46	5.4	5.14	2.16		20.21	19.96	29.2	35.62	37.96
***T. arvicolae***	32.66	19.85	19.84	20.1	20.21		17.72	28.34	35.17	37.34
***T. muris***	33.14	19.92	20.01	19.94	19.96	17.72		28.99	35.55	37.42
***T. cossoni***	34.48	29.21	29.06	29.28	29.2	28.34	28.99		37.2	38.48
***T. ovis***	36.8	35.46	35.55	35.65	35.62	35.17	35.55	37.2		36.72
***T. trichiura***	38.52	37.61	37.8	37.96	37.96	37.34	37.42	38.48	36.72	

## Data Availability

The data presented in this study are openly available in GenBank database (GenBank accession nos. MZ222133-MZ222134 and MZ229684-MZ229690).

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
