# Peer review of "Mitogenomics and Evolutionary History of Rodent Whipworms (Trichuris spp.) Originating from Three Biogeographic Regions"

_life, 2021, doi:10.3390/life11060540_

Round 1

Reviewer 1 Report

This article entitled 'Mitogenomics and evolutionary history of rodent whipworms (Trichuris spp.) originating from three biogeographic regions' was well designed and conducted analytically considering many aspects with correlated studies. It is very impressive that this study not only obtained sequence, but also conducted comparative analysis with pre-existing dataset.  It can be considered a reliable study that has been associated with each host species. However, several minor revisions needed to considert were followed by.

  • Line 18. T. suis and trichiura change to Trichuris suis and Trichuris trichiura
  • Line 19. It must to be actualized, different authors carried out several studies based on mitochondrials genes in order to infer the phylogenetic studies of Trichuris, even Trichuris spp. from rodents.
  • Line 22. cossoni, T. arvicolae and T. mastomysi change to Trichuris cossoni, Trichuris arvicolae and Trichuris mastomysi.
  • Line 56 Mus musculus in italics
  • Line 76. Include other Trichuris species such as pardinasi, T. bainae.
  • Line 102 change sp. to spp.
  • Table 1 Trichuris must be in italics and Gb. AN must be included for publication.
  • Line 129 cytochrome b in italics
  • Line 166. Indicate the program for substitution model. Is the same program for all genes analysed?. It must be stated and must be justified.
  • The results should be more up to date. Missing other species whose mitochondrial genome is available in Genbank for phylogenetics analysis ( suis. T. discolor, T. rinophiteroxella, Trichuris sp. from Papio papio. )
  • GTR + G was often favored when Model Test was given the option. This would seem to be the most parameter-rich option.  Any ideas as to why?

Author Response

Response to reviewer 1

This article entitled 'Mitogenomics and evolutionary history of rodent whipworms (Trichuris spp.) originating from three biogeographic regions' was well designed and conducted analytically considering many aspects with correlated studies. It is very impressive that this study not only obtained sequence, but also conducted comparative analysis with pre-existing dataset.  It can be considered a reliable study that has been associated with each host species. However, several minor revisions needed to considert were followed by.

We thank the reviewer for generally positive review of our manuscript. The reviewer's comments are in bold, our response is in default font.

Line 18. T. suis and trichiura change to Trichuris suis and Trichuris trichiura

Changed as requested

Line 19. It must to be actualized, different authors carried out several studies based on mitochondrials genes in order to infer the phylogenetic studies of Trichuris, even Trichuris spp. from rodents.

Changed as requested - we deleted the outdated sequence and replaced it with a more appropriate one. Line 21 now includes the statement: "despite an increasing demand for whole genome data" , instead of the outdated one.

Line 22. cossoni, T. arvicolae and T. mastomysi change to Trichuris cossoni, Trichuris arvicolae and Trichuris mastomysi.

Changed as requested

Line 56 Mus musculus in italics

Changed as requested

Line 76. Include other Trichuris species such as pardinasi, T. bainae.

Rejected - The line in question discusses those rodent Trichuris species for which there exist published records of whole mitogenome, not only mitochondrial markers. To our knowledge, there is no publication which describes the whole mitogenomes of such species and there is no Genbank record available. The only records on Genbank pertaining to these species are sequences of COX1, CYTB and ITS2 genes.

Line 102 change sp. to spp.

Changed as requested

Table 1 Trichuris must be in italics and Gb. AN must be included for publication.

Changed as requested - at the time we first submitted our paper, the GenBank had not yet provided us with the accession numbers. We have provided all ANs for both Trichuris and hosts in Table 1.

Line 129 cytochrome b in italics

Changed as requested

Line 166. Indicate the program for substitution model. Is the same program for all genes analysed?. It must be stated and must be justified.

Changed as requested - we have used jModelTest to test the best substitution models. We have included this information in the revised manuscript. Line 194 now includes: "jModelTest v2.1.10 [45] was used to test the substitution models."

The results should be more up to date. Missing other species whose mitochondrial genome is available in Genbank for phylogenetics analysis ( suis. T. discolor, T. rinophiteroxella, Trichuris sp. from Papio papio. )

Rejected - our analysis is only concerned with phylogeny of Trichuris from rodent hosts. The samples from T. ovis and T. trichiura were only included as another outgroup samples together with Trichinella spiralis and do not figure in any other analysis. Performing a phylogenetic analysis for every published whipworm mitogenome would vastly exceed the scope of the paper. In order to be more clear, we changed the lines 191-194 to:

"For the purpose of phylogenetic analysis, the mitogenomes of T. ovis (NC_018597), T. trichiura (NC_017750) and Trichinella spiralis (NC_002681) were added to the alignment and used as outgroup samples."

GTR + G was often favored when Model Test was given the option. This would seem to be the most parameter-rich option.  Any ideas as to why?

We suppose that this is likely a feature of any sufficiently big alignment as the GTR+G is the most robust substitution model and hence appropriate for any large dataset that includes deep divergences. However, the discussion of substitution models is beyond the scope of our paper.

Reviewer 2 Report

The manuscript recorded seven mitochondrial genomes of Trichuris species and investigated the phylogeny of the genus. This study increases our understanding of the phylogenetic relationships of Trichuris species. However, the findings were superficially provided and not described in deep. Authors are recommended to improve the manuscript before further consideration by the editor. Detailed comments are listed below.  

- Title: “Mitogenomics and evolutionary history of rodent whipworms (Trichuris spp.) originating from three biogeographic regions”. Even though this study performed comparative analysis of mitogenomes and analyzed the phylogeny of Trichuris species from different localities, it is not related to evolutionary history. Authors should revise the title to fit the content of the study.

- Introduction: “All the mitogenomes are circular, have the same organisation, and consist of 13 protein coding and 22 tRNA genes.” I recommend authors add the number of rRNA gene here. 

- Table 1: Why author did not provide Genbank accession numbers? It is compulsory to provide the numbers in this table.

- Materials and methods: Authors should add names of cities and countries for kits used in this study.

- Phylogenetic analysis: Why did authors only use Bayesian inference to build the phylogenetic tree? I strongly recommend authors use Maximum likelihood method, as well. From this finding, authors can compare the level of consistency between the two phylogenetic methods. Additionally, authors should describe more about phylogenetic analysis, such as what part of mitogenome was used for analysis (Protein-coding genes or amino acids...) and what software was used to determine the best-fit model?

- Non-coding region: Even though only two out of seven mitogenomes are complete, authors should not skip the non-coding region. I recommend author add information about the non-coding region of the mitogenome, especially for two complete mitogenomes, and compare them with previous records of species in the same genus.

- Authors should use a comma to separate thousands and hundreds units in nucleotide numbers in Table 2, for example: 14,003 instead of 14003.

Author Response

Response to reviewer 2

The manuscript recorded seven mitochondrial genomes of Trichuris species and investigated the phylogeny of the genus. This study increases our understanding of the phylogenetic relationships of Trichuris species. However, the findings were superficially provided and not described in deep. Authors are recommended to improve the manuscript before further consideration by the editor. Detailed comments are listed below.  

We thank the reviewer for the evaluation of our manuscripts and constructive comments to improve it. The reviewer's comments are in bold, our responses are in default font.

Title: “Mitogenomics and evolutionary history of rodent whipworms (Trichuris spp.) originating from three biogeographic regions”. Even though this study performed comparative analysis of mitogenomes and analyzed the phylogeny of Trichuris species from different localities, it is not related to evolutionary history. Authors should revise the title to fit the content of the study.

Rejected - We disagree with the reviewer. Performing a phylogenetic analysis allows us to estimate the lines of evolutionary descent of different species from a common ancestor, the order in which they branch, and the relative time intervals involved.  This is the “evolutionary history” we refer to in our title.

Introduction: “All the mitogenomes are circular, have the same organisation, and consist of 13 protein coding and 22 tRNA genes.” I recommend authors add the number of rRNA gene here. 

Changed as requested - Lines 29-31: "All the mitogenomes are circular, have the same organisation, and consist of 13 protein-coding, 2 rRNA genes and 22 tRNA genes".

Table 1: Why author did not provide Genbank accession numbers? It is compulsory to provide the numbers in this table.

Changed as requested - at the time we first submitted our paper, the GenBank had not yet provided us with the accession numbers. We have provided all ANs for both Trichuris and hosts in Table 1.

Materials and methods: Authors should add names of cities and countries for kits used in this study.

Changed as requested - we added the names of cities and countries for kits used on lines 126-137

Phylogenetic analysis: Why did authors only use Bayesian inference to build the phylogenetic tree? I strongly recommend authors use Maximum likelihood method, as well. From this finding, authors can compare the level of consistency between the two phylogenetic methods...

Changed as requested - We agree that Maximum likelihood analysis should be included together with Bayesian inference and the lack of it represented a weakness in our original manuscript. We have performed the ML analysis in the revised version and accordingly made changes to the materials and methods and results. Since the topology was the same in both analyses, we added the ML bootstrap values to the Figure 5.

The topology was the same in both ML and BI trees. To further compare the levels of consistency between the two methods used, we plotted the pairwise distance between each sample pair given by one method against the pairwise distances given by the second method. The results revealed a remarkable degree of similarity between the two methods and were summarised in a new supplementary figure (Figure S2, original S2 now being Figure S3).

We made the following additions to the manuscript:

Lines 195-198: "Phylogenetic analysis was performed via Bayesian inference (BI) conducted in MrBayes v3.2.6 [46] and complemented by a Maximum Likelihood phylogenetic analysis conducted in RaxML v8.2.12 [47]. The whole genome alignments were used for the phylogenetic analyses."

Lines 202-209: "To compare the level of consistency between the two phylogenetic methods, we first checked the topologies for any inconsistencies and then plotted the individual values of pairwise differences between same sample pairs given by the different methods against each other. This comparison was performed to compare the difference between branch lengths as given by the two methods of phylogenetic analysis. Only the Bayesian Inference analysis using MrBayes was performed for the host samples."

Lines 331-336: "Both Maximum Likelihood and Bayesian inference analyses gave the same phylogenetic topology (Figure 4) and very similar branch lengths. The difference between individual pairwise divergences for each method of phylogenetic analysis was low, ranging from 0 - 0.25, with average difference of 0.056 (for more details, see Figure S2). The phylogenetic analyses"

Line 384: "followed by ML bootstrap values."

Lines 509-513: "Figure S2: Pairwise comparison of branch lengths between each sample pair as constructed by the Maximum Likelihood and Bayesian inference phylogenetic analyses. The black line denotes equality, the red dashed line is the regression line from the linear model. The regression slope is equal to 1.06, while the intercept is equal to -0.05.; Figure S3: Phylogenetic tree of rodent hosts based on CYTB alignment. The node labels denote Bayesian Posterior Probabilities (BPP)."

Additionally, authors should describe more about phylogenetic analysis, such as what part of mitogenome was used for analysis (Protein-coding genes or amino acids...) and what software was used to determine the best-fit model?

Changed as requested - we used whole genome alignments for the phylogenetic analyses without partitioning. The software used for model fit was jModelTest v2.1.10

Changes to the manuscript:

Lines 194-198: "jModelTest v2.1.10 [45] was used to test the best scoring substitution models. Phylogenetic analysis was performed via Bayesian inference (BI) conducted in MrBayes v3.2.6 [46] and complemented by a Maximum Likelihood phylogenetic analysis conducted in RaxML v8.2.12 [47]. The whole genome alignments were used for the phylogenetic analyses."

Non-coding region: Even though only two out of seven mitogenomes are complete, authors should not skip the non-coding region. I recommend author add information about the non-coding region of the mitogenome, especially for two complete mitogenomes, and compare them with previous records of species in the same genus.

Changed as requested - we have added the summary statistics about NCR-L and NCR-S into Table 3 and commented on them in results, especially about NCR-L. We compared our records with the published genome of closely related Trichuris muris, which we deemed an appropriate sample for comparison, since it is also a rodent whipworm and we used it in our analyses elsewhere in the paper.

Changes to the manuscript:

Lines 260-266: "There were no differences in tRNA anticodons across the samples. The position of NCR-L could be inferred from the assembly even in those samples, being between the genes NAD1 and tRNA-Lys. However, very limited conclusions about the base composition in said region could be drawn, since the coverage of NCR-L was either too low or non-existent in the five samples mentioned. The NCR-S was located between NAD3 and tRNA-SerUCN."

Lines 291-298: "Both non-coding regions were heavily AT biased (AT percentages 78.6 % for NCR-L and 80.2 % for NCR-S). Although the information about NCR-L came from two samples only, these were very similar to the NCR-L of T. muris both in length (82 bp in T. sp. ex Mus caroli, 100 bp in T. sp. ex Mus mahomet vs. 112 bp in T. muris) and composition (GC content 21.4 % average in our samples vs 19.6 % in T. muris). The average length of NCR-S (107.4 bp in our samples vs. 106 bp in T. muris) and GC content was similar to the T. muris (19.8 % average in our samples vs. 20.8 % in T. muris)."

Authors should use a comma to separate thousands and hundreds units in nucleotide numbers in Table 2, for example: 14,003 instead of 14003.

Changed as requested

Reviewer 3 Report

The authors presented a very interesting manuscript contributing to add genetic knowledge on these parasites. I have only minor concerns. 

Nomenclature: I suggest to consider to change the nomenclature used for the isolates of Trichuris in figures. For example: T. ex mus caroli should be named as Trichuris sp. if the Trichuris species is not known. Maybe authors could assign a code to let the reader trace the host identity (Tsp_MC for example). This is just a suggestion and the authors used this nomenclature in table 1.

Abstract

we sequenced the mitogenomes of seven rodent hosts, belonging to three biogeographic re- 20 gions (Palearctic, Afrotropical and Indomalayan), including three previously described 21 species, T. cossoni , T. arvicolae and T. mastomysi . We assembled and annotated 2 complete

introduction

line36 delete (Trichuris trichiura),

line56-57 rodent species names in italics

line 65 pigs: T. trichiura error?

Author Response

Response to reviewer 3

The authors presented a very interesting manuscript contributing to add genetic knowledge on these parasites. I have only minor concerns. 

We thank the reviewer for positive evaluation of our manuscript. The reviewer's comments are in bold, our comments are in default font.

Nomenclature: I suggest to consider to change the nomenclature used for the isolates of Trichuris in figures. For example: T. ex mus caroli should be named as Trichuris sp. if the Trichuris species is not known. Maybe authors could assign a code to let the reader trace the host identity (Tsp_MC for example). This is just a suggestion and the authors used this nomenclature in table 1.

Partially accepted - We agree that unidentified species should be marked as Trichuris sp. so we added the sp. where it was missing through the text and figures. We considered changing the nomenclature to special codes but ultimately decided to stay with our original albeit improved nomenclature as we believe that from the text both the host species and status of the Trichuris sample (described vs. non-described species) are sufficiently clear. We also believe that in case of e.g. Figure 4., including the whole names of host species is more clear than any other labelling method as it allows the reader to immediately identify the hosts without having to look up the labels.

we sequenced the mitogenomes of seven rodent hosts, belonging to three biogeographic re- 20 gions (Palearctic, Afrotropical and Indomalayan), including three previously described 21 species, T. cossoni , T. arvicolae and T. mastomysi . We assembled and annotated 2 complete

We did not understand what the reviewer is suggesting here.

line36 delete (Trichuris trichiura),

Changed as requested

line56-57 rodent species names in italics

Changed as requested

line 65 pigs: T. trichiura error?

Changed to T. suis, this was a simple labelling error.

Round 2

Reviewer 2 Report

The authors revised the manuscript well according to previous comments.
So, it can be accepted as current form.

Author Response

We thank the reviewer for evaluation of our manuscript